# Effects of Fish Skin Gelatin Hydrolysates Treated with Alcalase and Savinase on Frozen Dough and Bread Quality

**DOI:** 10.3390/foods13010139

**Published:** 2023-12-30

**Authors:** Sefik Tekle, Gorkem Ozulku, Hatice Bekiroglu, Osman Sagdic

**Affiliations:** 1Department of Food Processing, Kaman Vocational School, Kirsehir Ahi Evran University, Kirsehir 40100, Turkey; sefiktekle@ahievran.edu.tr; 2Department of Food Engineering, Faculty of Chemical and Metallurgical Engineering, Yildiz Technical University, Istanbul 34220, Turkey; ozulkug@yildiz.edu.tr; 3Food Engineering Department, Agricultural Faculty, Sirnak University, Sirnak 73300, Turkey

**Keywords:** sea bream gelatin, enzymatic hydrolysis, frozen storage of bread dough, bread quality

## Abstract

Fish skin gelatin, as a waste product of sea bream, was used to obtain fish gelatin hydrolysate (FGH) with the treatment of alcalase (alc) and savinase (sav). The functional properties of FGHs and their usage possibilities in frozen dough bread making were investigated. FGH treated with alc showed a higher emulsifying stability index (189 min), while FGH treated with sav showed greater foaming capacity (27.8%) and fat-binding capacity (1.84 mL/g). Bread doughs were produced using two FGHs (alc and sav) and their combination (FGH-alc + FGH-sav). Using FGH treated with these enzymes individually was more effective than their combination in terms of polyacrylamide gel electrophoresis (SDS-PAGE) results and bread quality (specific volume and hardness). The addition of FGH into bread dough showed no significant effect on bread dough viscoelasticity (tan δ), while the increment level of tan δ value for control dough was higher than the dough containing FGH after frozen storage (−30 °C for 30 days). The highest freezable water content (FW%) was found in control dough (33.9%) (*p* < 0.05). The highest specific volume was obtained for control fresh bread and bread with FGH-alc, while the lowest volume was obtained for fresh bread containing FGH-sav (*p* < 0.05). After frozen storage of the doughs, the bread with FGH-alc showed the highest specific volume. FGH addition caused a significant reduction in the L* (lightness) value of fresh bread samples when compared to control bread (*p* < 0.05). This study suggested that usage of FGH-alc in bread making decreased the deterioration effect of frozen storage in terms of the specific volume and hardness of bread.

## 1. Introduction

Bread products with high volume, a soft and elastic crumb, a brittle and golden-brown crust, and long shelf life have been the consumer expectation from the bakery industry for years. Bread is known to have one of the oldest production process [1]. Although there are many different bread production techniques resulting from access to raw materials as well as regional and cultural differences, some basic problems are encountered such as retrogradation caused by physicochemical changes in the structure of starch during storage, hardness increment, and premature staling due to increased moisture loss and water mobility [2]. All these changes cause a loss of quality and the bread’s unique aroma and taste, thus decreasing consumer acceptability [3,4].

More efficient techniques, such as frozen dough bread making, have been developing for years in order to offer the opportunity of fresh bread at any time. With the use of frozen dough, eliminating the problems that result in loss of quality in conventional baking can be achieved, especially at critical points such as storage and shipment. Also, frozen dough bread making provides a standard end product, enables large-scale and industrial production, and saves time [1,5,6]. On the other hand, there are some disadvantages caused by the freezing process and temperature fluctuation during storage in frozen dough bread making. The water migration from gluten leads to the formation of ice crystals, damaging the gluten network and yeast cells. Therefore, a decrease in yeast activity and weakening of the dough strength are mainly observed, resulting in the poor baking performance of frozen dough [7,8]. To overcome such problems, several methods for the frozen dough production process have been studied. Zhang et al. [9] categorized these methods as the selection and optimization of raw materials, the improvement of yeast freeze tolerance, the incorporation of additives, and the amelioration of freezing, storage, thawing, and dough-making processes.

Nowadays, the increasing demand for functional foods with rich and healthy nutritional content has brought about the need to make frequently consumed foods such as bread more functional. In this context, protein and protein hydrolysates are among the important food additives and are widely used in the improvement of final products in terms bioactive properties, structural, and sensorial characteristics [10,11]. In particular, the production of protein and protein hydrolysate from waste and food industry by-products has become very important within the context of sustainability. Enzymatic hydrolysis treatment enables the conversion of protein-rich fish by-products, which have a large waste potential, into high value-added protein hydrolysate. Fish waste, which has a crude protein content of 8–35% depending on the type of fish, is a potential source of polyunsaturated fatty acids, essential amino acids, enzymes, gelatin, and collagen [12,13,14]. Gelatin, found in large amounts in fish waste such as skin and bone, is a soluble protein that can be produced via the partial denaturation of collagen in connective tissue and has wide applicability [15,16]. Gelatin hydrolysates, which have a much smaller molecular weight peptide composition, exhibit high bioactive and technological functions such as emulsifying, foaming, and colloidal stabilizer properties. These functions vary depending on the degree of enzymatic hydrolysis and therefore the molecular weight and composition of the peptides. Confectionery, meat, dairy, and bakery products provide the desired structural and textural properties by enriching gelatin hydrolysates [16,17,18].

The usage of protein hydrolysates in frozen dough studies has been widely investigated in order to minimize the quality deterioration of frozen dough bread. It has been reported that the damaging of the gluten network and the decreasing yeast activity were slowed down in the studies using different protein hydrolysates such as milk [19,20], sweet potato protein [7], pig skin gelatin [21], bovine collagen [22], and silver carp hydrolysates [23]. Moreover, Sang et al. [24] suggested that the use of fish skin gelatin in bread making improved the strength and gas-retention capacity of dough due to its having good gelation and emulsification properties. Therefore, in this study, we aimed to produce high value-added fish gelatin hydrolysate (FGH), obtained by applying alc and sav (serine type enzymes) to fish gelatin produced from sea bream skin waste, and to investigate some of its technological properties, as well as its usage possibilities in making bread from frozen dough.

## 2. Materials and Methods

### 2.1. Materials

Alcalase^®^ (protease from *Aspergillus oryzae*), savinase (protease from *Bacillus* sp., 16 U/g), and phosphate-buffered saline (PBS) were purchased from Sigma-Aldrich Chemie GmbH (Taufkirchen, Germany). Acrylamide, sodium dodecyl sulfate (SDS), and the other chemicals were provided by Sigma-Aldrich (St. Louis, MO, USA). The protein molecular mass marker PageRuler^TM^ (prestained protein ladder 10–180 kDa), used to detect different molecular weights in the bands for SDS page analysis, was purchased from Thermo Fisher Scientific (Waltham, MA, USA).

Dry yeast (Pakmaya), salt (Billur), and wheat flour were purchased from a local market for bread making. The wheat flour in this study contains 10.8% protein, 0.73% ash, and 13.5% moisture.

### 2.2. Productions of Fish Skin Gelatin and Its Hydrolysate and FGH Characteristics

#### 2.2.1. Production of Fish Skin Gelatin

Sea bream fish skins were provided by Kilic Deniz A.S (Mugla/Turkey). Fish skins obtained as a result of the fillet process were delivered to the laboratory in packages containing ice molds. The skins containing no scales and meat residues were washed with cold tap water, and the water was removed. The obtained skins, which were kept in the freezer at −20 °C (less than 2 months) until processed, were thawed in the refrigerator (+4 °C) for gelatin production and then cut into pieces (~2–3 cm^2^ sizes).

Gelatin production was carried out using the method specified by Boran and Regenstein [25]. Fish skins were first treated with NaOH (5:1 *v*/*w*, 0.55 N, 67.5 min) alkaline solution and then with HCl (5:1 *v*/*w*, 0.1 N, 45 min) acid solution. After each treatment, distilled water (5:1 *v*/*w*) was used for washing the skins. Afterwards, extraction was carried out with pure water (4:1 *v*/*w*, 50 °C, 3 h) in a water bath (Memmert WNB45, Schwabach, Germany). The obtained gelatin solutions were put into glass containers and the drying process was performed (60 °C, ~72 h). The final moisture content of the produced gelatin was 5.48% after the drying process. The gelatin leaves were stored in polyethylene packages in a dry environment until use.

#### 2.2.2. Production of the FGHs

A 5% (*w*/*v*) solution was prepared from gelatin with pure water and dissolved homogeneously. Enzymatic hydrolysis was carried out with alc and sav, and its pH was adjusted to 8.0 (60 °C) and 9.0 (50 °C), respectively, with 0.1 M NaOH/HCl solution (E/S:2.5/100 [*v*/*p*]) for 3 h. After hydrolysis, the enzymes were inactivated by heat (90 °C, 20 min), and the solutions were centrifuged at 10,000 rpm for 15 min (Andreas Hettich GmbH and Co. KG, Tuttlingen, Germany). The resulting gelatin hydrolysates were dried with a freeze dryer (Martin Christ GmbH, Beta 1–8 LSCplus, Osterode am Harz, Germany), packaged, and stored.

#### 2.2.3. Determination of the Degree of Hydrolysis (DH) of the FGHs

The degree of hydrolysis of FGH was determined via the pH stat method [26]. The degree of hydrolysis is expressed as the ratio of the peptide bonds (h) broken as a result of hydrolysis to the total number of bonds per unit weight (h_tot_). The degree of hydrolysis is calculated according to the following Equation (1):%DH = h × 100/h_tot_ = (B × N_b_ × 100)/(α × Mp × h_tot_),(1)
where B, Nb, α, and Mp are base amount, base normality, mean dissociation constant of the α-NH_2_ groups, and the amount of protein (g), respectively.

The total peptide bonds refers as h_tot_ (meqv/g protein) = 8.41 mmol/g protein [27].

#### 2.2.4. Zeta Potential of the FGHs

A 10% solution of FGH was prepared with pure water and completely dissolved, and 25 μL of the sample and 2 mL of PBS (0.01 M and pH 7.4) were mixed and measured with Zetasizer Nano ZSP (Malvern, UK). Values were averaged from 3 studies with at least 15 measurements.

#### 2.2.5. Emulsifying Activity and Stability of the FGHs

The method of Zamorano-Apodaca et al. [28], with some modifications, was used to determine the emulsion activity index (EAI) and stability index (ESI) of FGH. A total of 30 mL of gelatin hydrolysate solutions (0.1% *p*/*v*), prepared with PBS (0.01 M and pH 7.4), were mixed with corn oil (3 mL). The homogenization with Ultra-turrax (Daihan HG-15D, Seoul, Korea) was performed at 18,000× *g* for 1 min, and 50 μL of sample was taken from the bottom of the emulsion and diluted to 5 mL with 0.1% SDS (sodium dodecyl sulphate) (*w*/*v*) at 0 and 10 min after homogenization. The absorbance of the samples was measured with a spectrophotometer (Shimadzu UV-1800, Tokyo, Japan) at 500 nm. The obtained absorbances (A_0_ and A_10_) were used in the following equations, and EAI (Equation (2)) and ESI (Equation (3)) values were calculated.
EAI (m^2^/g) = (2 × 2.303 × A_0_)/(C × φ)(2)

A_0_ = Absorbance at 500 nm wavelength;

C = Initial protein concentration (g/mL);

Φ = Oil volume in emulsification.
ESI (min) = (A_0_ × Δt)/ΔA(3)

A_10_ = Absorbance at the end of 10 min;

Δt = 10 min;

ΔA = A_0_ − A_10_.

#### 2.2.6. Foaming Capacity and Stability of the FGHs

The foam capacity and foam stability of FGHs were determined via the method of Zamorano-Apodaca et al. [28], with slight modifications. An amount of 10 mL of gelatin hydrolysate solution (0.5% *p*/*v*) was taken and homogenized with Ultra-turrax (Daihan HG-15D, Seoul, Korea) at 14,000× *g* for 1 min. After homogenization, the total foam volume was measured at time 0-, 3-, and 10-min. Foam expansion at 0 min refers to foam capacity, while foam expansion at 10 min refers to foam stability. Foam expansion was calculated by the following Equation (4):Foaming expansion (%) = [(A − B)/B] × 100.(4)

A: Volume at different times (mL);

B: Volume before homogenization (mL).

#### 2.2.7. Fat-Binding Capacity of FGH

The fat-binding capacity of FSH was determined according to the method of Karoud et al. [29]. A 50 mg sample was placed in tare centrifuge tubes, and 1 mL of sunflower oil was added and left at room temperature for 1 h. The mixtures were vortexed (Velp ZX3 Vortex Mixer, Usmate Velate, Italy) for 5 sec every 15 min. The samples were then centrifuged at 5000 rpm for 20 min (Andreas Hettich GmbH and Co. KG, Tuttlingen, Germany) and the oil phase was removed. For the control sample, only oil was added into an empty centrifuge tube. Fat-binding capacity was expressed in mL of fat/g protein.

### 2.3. Preparation of Fresh/Frozen Bread Dough and Baking

The breadmaking method of this study was based on our previous study [19]. The basic recipe for fresh and frozen dough included 100 g wheat flour, 2 g dry yeast, 1.5 g salt, and 58 g water. The water absorption value of wheat flour was determined according to the AACC method 54-21.02 [30]. The bread doughs containing fish gelatin hydrolysates (FGH) produced with alc and sav were prepared by replacing wheat flour with FGH at the level of 1%. The combination of both FGHs (alc+sav) was also incorporated into bread dough, individually, at the level 0.5%. The kneading of all ingredients was performed in KitchenAid (Benton Harbor, MI, USA) at speed 4 for 5 min. Then, the doughs were shaped. The storage of frozen dough samples was performed at −30 °C for 30 days. After storage, the doughs were thawed at 4 °C for 3 h. The fermentation (30 °C and 85% relative humidity for 2 h) was carried out in a cabinet (Nuve TK 252, Ankara, Turkey). An electrical oven (Maksan, Nevsehir, Turkey) was used to bake the dough (235 °C, 25 min).

#### 2.3.1. Rheological Properties of the Bread Doughs

The rheological properties of the bread doughs were determined by using a rheometer (Antonpaar MCR 302, Graz, Austria).

About 2 g of dough was placed on the parallel plate with a 2 mm gap, and 0.1% strain in the linear viscoelastic region was chosen for frequency sweep tests between 0.1 rad/s and 100 rad/s at 25 °C. Duplicate measurements were performed.

#### 2.3.2. Determination of Freezable Water Content of the Bread Doughs

A differential scanning calorimeter (DSC; TA instrument Q20, New Castle, DE, USA) was used for the determination of freezable water (FW) content according to the method of Chen et al. [7]. FW content was calculated using Equation (5):FW (%) = ΔH/ΔH_o_ × W_c_ × 100.(5)

∆H: The enthalpy (J/g) of the melting peak of the endothermic curve;

∆H_o_: The enthalpy (334 J/g) of melting peak of pure water;

W_c_: The total water content (%) of the dough.

The total water content was determined via a moisture analyzer (Radwag MR50, Poznań, Poland). The analysis was carried out with 2 replications for each sample.

#### 2.3.3. Sodium Dodecyl Sulfate–Polyacrylamide Gel Electrophoresis (SDS-PAGE) Analysis of the Bread Doughs

The method of Laemmli [31] was used for SDS-PAGE analyses of all the bread dough samples. Protein extraction of the freeze-dried and powdered dough samples was carried out according to the method described by Yu et al. [21]. Protein fractions of the dough samples were collected via extraction completed using a buffer of pH 6.8 (0.125 mol/L Tris-HCl, 10% (*v*/*v*) glycerol, 2% (*w/v*) sodium dodecyl sulphate, 0.01% *w*/*v* bromophenol blue). Collected protein samples were mixed with sample buffer at the ratio of 4:1 (*v*:*v*) and incubated in a boiling water bath (5 min). The 20 µL samples were added to each loading well. The electrophoresis was run under a 20 mA/gel electric current in the separation gel (7.5% and 20% acrylamide) in a vertical electrophoresis system (Mini-PROTEAN^®^ System, Bio-Rad, Hercules, CA, USA). After the electrophoresis, the bands were exposed to a staining solution (Coomassie brilliant blue:methanol:acetic; acid:water; 0.25%:50%:10%:39.75%). The excess dye was removed with the destaining solution (methanol:acetic; acid:water; 50%:10%:40%). Then, the bands were visualized with the Biorad Gel Doc EZ imaging system.

#### 2.3.4. Secondary Structure Measurement of Proteins by FTIR of the Bread Doughs

Lyophilized dough samples were compressed and spectrally scanned by the FTIR spectrometer equipped with a DLa TGS detector (Bruker Tensor 27, Germany). FT-IR spectra of all samples were obtained with wave numbers from 400 to 4000 cm^−1^ during 16 scans per spectra with 2 cm^−1^ resolution to determine the secondary structure of protein. Deconvolution was performed on all spectra via the interpretation of changes in the overlapping amide I band (1600–1700 cm^−1^) using Origin 2020b software [32]. Three measurements were conducted for each sample.

#### 2.3.5. Determination of Specific Volume, Texture and Color Characteristics of the Breads

The specific volumes of the bread samples were calculated by dividing the volume of bread (mL) by the weight (g) of bread. The volume of breads was determined according to AACC [30].

For texture profile analysis (TPA), a 5 kg load cell and a 36 mm diameter cylindrical compression probe were used (SMS TA.XT2 Plus, Glasgow, UK). The TPA test was conducted with 50% compression, 5.0 mm/s test speed, and 5 s delay time between the two compression cycles. Duplicate measurements were performed.

The lightness (L*), redness (a*), and yellowness (b*) values of the crust and crumb color of bread samples were measured using a Chromameter (CR-100 Konica Minolta, Tokyo, Japan). The following Equation (6) was used to calculate color differences (∆E*):ΔE* = [(ΔL*)^2^ + (Δa*)^2^ + (Δb*)^2^]^1/2^,(6)
where ΔL*, Δa*, and Δb* are the differences for L*, a*, and b* between the sample and control fresh bread, respectively.

### 2.4. Statistical Analysis

SPSS 19.0 software (Chicago, IL, USA) was used for statistical analysis of the results (mean ± standard deviation). An analysis of variance (ANOVA), followed by Duncan’s multiple range tests, was performed. A probability value of *p* < 0.05 was considered statistically significant.

## 3. Results

### 3.1. Some Characteristics of the FGHs

#### 3.1.1. Degree of Hydrolysis of the FGHs

The degrees of hydrolysis of FGHs are given in Figure 1. The degree of hydrolysis also increased as the enzymatic hydrolysis time increased. The degree of hydrolysis was determined between 9.37 and 19.36% for FGH treated with alc, while it was between 7.54 and 17.44% for FGH treated with sav. At the end of three hours of hydrolysis, the hydrolysis degree of FGH treated with alc was determined as 19.36%, and the hydrolysis degree of FGH treated with sav was determined as 17.44%. The difference in the degree of hydrolysis could be due to the enzyme type and specificity [11]. Alc and sav are serine-type endoproteases with broad substrate specificity. The actions of the two enzymes were at similar levels in the production of FGH (Figure 1).

#### 3.1.2. Zeta Potential of the FGHs

Zeta potential is the electrical potential between the particle surface and the dispersion liquid in colloidal systems, varying with the distance from the particle surface, and it also expresses the prevalence of negative charge in hydrolysates [11,18]. It has been stated that emulsions with high zeta potential are more electrostatically stable and therefore have better physical properties [11]. In this study, the zeta potential of FGH treated with alc and sav was found to be −27.28 ± 0.98 and −21.36 ± 0.24, respectively. The differences in zeta potential were observed depending on the enzyme type. FGH treated with alc showed higher zeta potential than FGH treated with sav (*p* < 0.05, Table 1). These results indicated that more carboxylate ions (COO−) were formed as a result of enzymatic hydrolysis [33].

#### 3.1.3. Emulsifying Activity and Stability of the FGHs

The emulsifying activity index (EAI) and emulsifying stability index (ESI) values of FGH treated with different enzymes (alc and sav) are given in Table 1. While the EAI value of FGH treated with sav was higher that of than FGH treated with alc, the opposite trend was shown for ESI value of the samples (Table 1). Enzyme type caused significant differences in both EAI and ESI (*p* < 0.05). FGH treated with sav with a lower degree of hydrolysis had a higher EAI value. As the degree of hydrolysis increases, low-molecular-weight peptides were formed, and these peptides were rapidly adsorbed to the oil/water interface, but they cannot remain stable for a long time because they cannot be adequately opened and rearranged there. On the contrary, peptides with higher molecular weights form more stable emulsions since showing these properties [34]. In addition, FGH treated with alc, which had a higher zeta potential value, showed a higher ESI value.

#### 3.1.4. Foaming Capacity and Stability of the FGHs

The foaming capacity and stability of gelatin hydrolysates are given in Table 1. Both the foaming capacity and stability of FGH treated with sav were found to be higher (*p* < 0.05). This showed that the enzyme type was important with respect to this property. In addition, the degree of hydrolysis could also affect the foam properties. FGH with lower a hydrolysis degree was found to have high foam capacity and stability (Table 1). It was stated that proteins/peptides with low molecular weight cannot rearrange their structure at the air–water interface and therefore cannot demonstrate the required power for foaming [35].

#### 3.1.5. Fat-Binding Capacity of the FGHs

Fat-binding capacity is one of the most important technological properties for the food industry and reveals the ability of proteins to interact with lipids [36]. The fat-binding capacity of FGH treated with the sav enzymes was approximately two times higher than that treated with alc (*p* < 0.05, Table 1). The enzyme type caused a significant difference in the fat-binding capacity. It has been reported that hydrolysates with a lower degree of hydrolysis exhibit higher fat-binding capacity [11]. Similar results were obtained in this study since the fat-binding capacity of FGH treated with sav was found to be higher due to its lower degree of hydrolysis.

### 3.2. Some Characteristics of the Bread Dough Produced by Adding FGHs

#### 3.2.1. Changes in the Viscoelastic Behavior of the Bread Doughs

As a result of dynamic rheological properties, the tan δ (G″/G′) values of both fresh and frozen dough samples are shown in Figure 2. Tan δ was considered an important indicator of the degree of viscoelasticity in many frozen dough studies. A tan δ value which is lower than 1 indicates that elastic behavior is more dominant than viscous behavior [37]. Frozen storage causes a rise in the tan δ value of dough samples (Figure 2), which shows the increment in the viscous properties of dough and the weakening of the gluten structure, as reported in many studies [1,7,38]. According to the study of Xin et al. [8], the gluten network damage was attributed to a progressive increase in ice crystals during the freezing process. After 30 days of frozen storage, the increment level of tan δ value for the control dough was 42.6%, which was higher than the dough containing FGH (Figure 2). This result shows that the addition of FGH could inhibit the damage to the gluten network by enhancing the polymer concentration in dough [8]. Similar results related to the improvement effect of protein hydrolysate addition, such as sweet potato protein hydrolysates [7], pigskin gelatin [21], and a desalted egg white and gelatin mixture system [38], to frozen dough have been reported.

#### 3.2.2. SDS-PAGEs of the Bread Doughs

The effects of FGHs treated with alc and sav on fresh and frozen dough samples were investigated using SDS-PAGE in order to monitor the depolymerization of proteins caused by frozen storage. The changes in the molecular weight of fresh dough samples produced with the addition of FGHs treated with alc, sav, and the combination of the two hydrolysates are presented in Figure 3A. The band intensity of fresh dough containing both FGH-alc and FGH-sav (FD-alc+sav) was lower than other fresh doughs (Figure 3A). This can be explained by the alteration of the aggregation between glutenin molecules and the formation of new protein polymers which were unable to enter the resolving gel due to their larger size [39,40]. The addition of FGH-alc and FGH-sav increased the band intensity in the α/β gliadin region for fresh dough samples when compared to control fresh dough (FD-control), showing possible new interactions between FGH, gluten, and water during dough formation [40]. The intensity of the bands in all frozen dough samples tended to increase when compared to the fresh dough samples (Figure 3B). The freezing process causes depolymerization in the protein matrix of wheat dough [41]. The depolymerization reaction in frozen dough samples was more notable in frozen control dough (FRD-control) and frozen dough containing both FGH-alc and FGH-sav (FRD-alc+sav) since the intensity of the bands in the ω gliadin, LMW-GS-γ gliadin, and α/β gliadin region increased (Figure 3B). This could be due to the recrystallization of ice and the redistribution of water during frozen storage caused the depolymerization of glutenin [42]. This effect caused by frozen storage was less notable in the bread doughs containing FGH-alc and FGH-sav individually (Figure 3), indicating a slowing down of the damage to the gluten network during frozen storage.

#### 3.2.3. Analysis of Secondary Structures of Protein of the Bread Doughs

In this study, the amide I region (1700–1600 cm^−1^), which expresses the secondary structures of gluten proteins, is given in Figure 4. In the amide I region, α-helices (1650–1660 cm^−1^) and β-sheets (1682–1696 cm^−1^) are more stable and regular, while β-turns (1662–1681 cm^−1^) and random coils (1640–1650 cm^−1^) represent irregular structures [43]. The decreasing level of α-helices structure caused by frozen storage was higher for dough samples containing the combination of the two FGHs (alc+sav) when compared to other samples (Figure 4). This result shows that addition of FGH individually could slow down the changes of α-helices’ structure by inhibiting the recrystallization of the ice crystals. A similar observation was reported by Bekiroglu et al. [19] and Zhou et al. [38], who used casein hydrolysate and gelatin mixture in frozen dough bread making, respectively. For fresh dough samples (FD), individual FGH addition increased β-turns structure but decreased the random coil structure when compared to control fresh dough (FD-control). Similar trends in terms of β-turns and random coil structure was observed after frozen storage (Figure 4).

#### 3.2.4. Freezable Water of the Bread Doughs

The melting onset temperature (Tm, onset), enthalpy (ΔH), and freezable water (FW) content of all samples are presented in Table 2. FW content is one of the important parameters used to determine the frozen dough quality since the formation of ice crystals is caused by freezable water [22]. FGH addition into dough formulation increased the Tm value on day 0. After frozen storage, the highest Tm value was obtained for FRD-control and FRD-sav samples. A lower Tm is desirable for economic advantages [44]. The FW content of the doughs increased due to temperature fluctuations during frozen storage, which cause the recrystallization of water [8]. FW content increased by 39% for control dough, 25% for dough containing FGH-alc, 26% for dough containing FGH-sav, and 9.4% for dough containing FGH-alc+sav, suggesting that the addition of FGHs could inhibit the conversion of bound water to freezable water. This also shows that specific interactions between free water and FGHs were formed, leading to the enhancement of the water-holding capacity of the gluten network [22,45].

### 3.3. Some Characteristics of the Breads from the Bread Dough with FGHs

The effect of FGH treated with alc and sav on the quality of breads made by frozen dough was investigated in terms of specific volume, hardness, and image characteristics (Figure 5). The highest specific volume was obtained for control bread (control-B), while the lowest one was Sav-B bread with FGH-sav (Sav-B) on day 0 (fresh bread). No significant effect was observed between control-B and bread and FGH-alc (Alc-B) samples in terms of the specific volume and hardness value on day 0 (Figure 5). However, the effects of FGH-sav and FGH-alc+sav on the specific volume and hardness value were similar on day 0. The usage of the combination of the FGH treated with alc and sav did not improve the quality characteristics of fresh breads in terms of specific volume and hardness. After frozen storage of dough samples (−30 °C for 30 days), the highest specific volume and lower hardness value were obtained for Alc-B and Sav-B samples, indicating that usage of these two FGHs cause less damage to bread quality. This result was also in line with the FW (%) content results (Table 2), since the doughs containing FGH-alc and FGH-sav revealed a lower increment level of FW, unlike the control dough. Lower FW (%) content indicates less damage to the gluten network caused by frozen storage [7]. Many studies have stated that the specific volume of breads made by frozen dough decreased and their hardness increased with the prolongation of frozen storage. Also, the degree of these changes could be controlled by using protein hydrolysate from different sources in frozen dough studies [7,21,38].

Table 3 presents the crust and crumb color of bread samples. The L* value of the fresh bread crust decreased due to the FGH addition (*p* < 0.05). The reduction in the L* value shows the browning of the bread crust. This can be attributed to intensive Maillard reaction products (compared to control bread) caused by the incorporation of FGH as a protein source. In a study which investigated the effect of casein hydrolysate on frozen bread, the browning effect was observed similarly [19]. FGH addition reduces the crust b* value of fresh breads, except for the bread containing both FGHs (Alc+Sav-B), but no significant effect was shown after frozen storage (Table 3). As for the color differences (ΔE), there were no differences between the crust color ΔE value of the fresh bread samples (*p* > 0.05), but the significant reduction was shown after frozen storage (*p* < 0.05). In terms of the ΔE value of bread crumbs, a lower ΔE value was observed in fresh bread containing FGH treated with alc, but there was no significant difference between bread samples after frozen storage (*p* > 0.05). The color differences are evaluated as very distinct, distinct, and small differences if the ΔE of the samples is higher than 3, between 1.5 and 3, and lower than 1.5, respectively [46]. According to this classification, the breads containing FGH had very distinct crust color differences based on the reference sample (fresh Control-B). After frozen storage, Alc-B and Sav-B showed distinct crust color difference, while Alc+Sav-B and Control-B had very distinct and small differences, respectively. This result shows that FGH addition led to a very perceivable difference in crust color compared to the effect of frozen storage. For crumb color differences, all bread samples had very noticeable color differences except for fresh Alc-B (Table 3).

## 4. Conclusions

Fish gelatin hydrolysates produced using alc and sav were evaluated in frozen dough bread making in order to slow down the quality deterioration of frozen storage (−30 °C for 30 days). The deterioration effect of frozen storage was in line with similar studies related to frozen dough bread. Control bread had a significant deterioration effect, showing higher freezable water content, a higher increase in tan δ, and lower bread making quality when compared to bread containing FGH. The results also indicated that the combination of FGH-alc and FGH-sav had no effect on improving the quality of frozen dough bread. The individual effect of the two FGHs on decreasing the deterioration caused by frozen storage in terms of bread specific volume and hardness was more drastic. In conclusion, FGH treated with alc was found to be more effective and could be considered as a beneficial additive for frozen dough bread making from both technological and nutritional perspectives.

## Figures and Tables

**Figure 1 foods-13-00139-f001:**
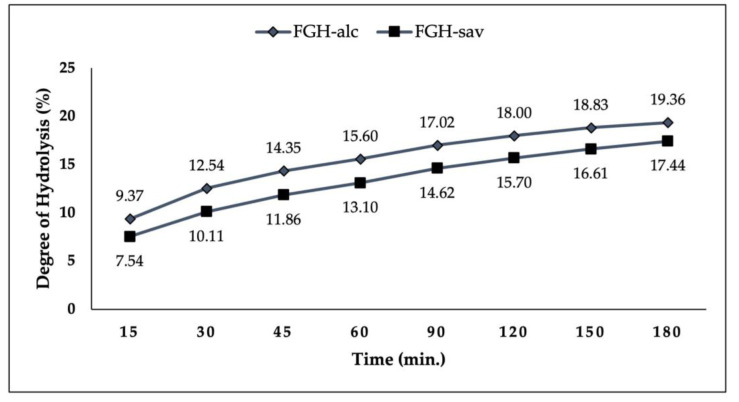
The degree of hydrolysis of gelatin treated with alcalase and savinase enzymes. FGH-alc: FGH treated with alcalase; FGH-sav: FGH treated with savinase.

**Figure 2 foods-13-00139-f002:**
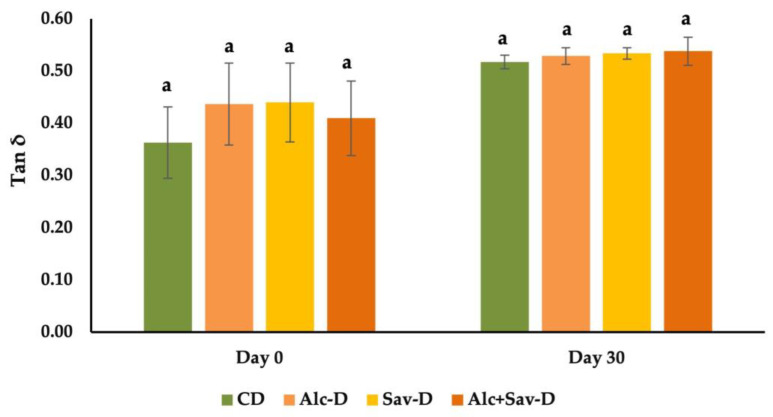
The effects of fish gelatin hydrolysates (FGHs) and frozen storage on the tan δ value of dough samples at a frequency of 1 Hz. CD: control dough; Alc-D: dough containing FGH treated with alcalase; Sav-D: dough containing FGH treated with savinase; Alc+Sav-D: dough containing both FGH. Same letters show no significant differences at the same frozen storage time.

**Figure 3 foods-13-00139-f003:**

(**A**) SDS-PAGE patterns of the fresh dough samples. FD-control: fresh control dough; FD-alc: fresh dough containing FGH treated with alcalase; FD-sav: fresh dough containing FGH treated with savinase; FD-alc+sav: fresh dough containing FGH treated with alcalase + FGH treated with savinase. (**B**) SDS-PAGE patterns of the frozen dough samples. FRD-control: frozen control dough; FRD-alc: frozen dough containing FGH treated with alcalase; FRD-sav: frozen dough containing FGH treated with savinase; FRD-alc+sav: frozen dough containing FGH treated with alcalase + FGH treated with savinase; FGH: fish gelatin hydrolysate; MW: the protein molecular marker PageRulerTM (prestained protein ladder 10–180 kDa).

**Figure 4 foods-13-00139-f004:**
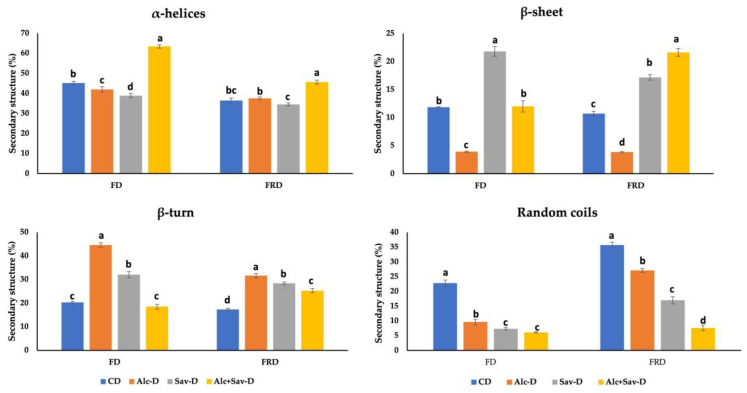
The effects of fish gelatin hydrolysates (FGHs) and frozen storage on the secondary structure of dough samples. CD: control dough; Alc-D: dough containing FGH treated with alcalase; Sav-D: dough containing FGH treated with savinase; Alc+Sav-D: dough containing both FGH. Same letters show no significant differences at the same frozen storage time (*p* > 0.05).

**Figure 5 foods-13-00139-f005:**
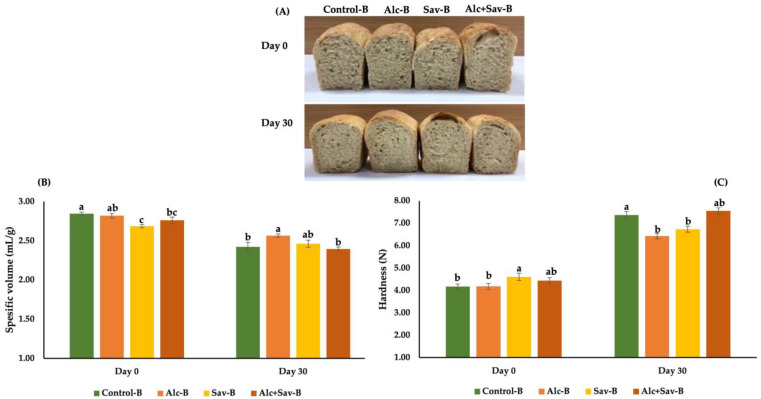
The effects of fish gelatin hydrolysates (FGHs) and frozen storage on bread physical properties (**A**), specific volume (**B**) and hardness (**C**). Control-B: control bread; Alc-B: bread containing FGH treated with alcalase; Sav-B: bread containing FGH treated with savinase; Alc+Sav-B: bread containing both FGH. Same letters show no significant differences at the same frozen storage time.

**Table 1 foods-13-00139-t001:** Physicochemical properties of the fish gelatin hydrolysates (FGH).

Enzyme	Zeta Potential (mV)	EAI (m^2^/g)	ESI (min)	Foam Capacity (%)	Foam Stability (%-3 min)	Foam Stability (%-10 min)	Fat-Binding Capacity (mL/g)
FGH-alc	−27.28 ± 0.98 ^a^	34.82 ± 0.9 ^a^	189.00 ± 1.00 ^a^	22.49 ± 1.50 ^a^	15.40 ± 0.79 ^a^	9.40 ± 0.79 ^a^	0.81 ± 0.06 ^a^
FGH-sav	−21.36 ± 0.24 ^b^	70.20 ± 0.85 ^b^	24.42 ± 0.62 ^b^	27.80 ± 2.69 ^b^	19.65 ± 1.00 ^b^	12.74 ± 0.79 ^b^	1.84 ± 0.03 ^b^

Data are expressed as mean ± standard deviation of triplicate determinations. Means with different lowercase letters in the same column indicate significant differences (*p* < 0.05) for the samples at the same storage time. FGH-alc: FGH treated with alcalase; FGH-sav: FGH treated with savinase.

**Table 2 foods-13-00139-t002:** Effect of the addition of fish gelatin hydrolysate (FGH) and frozen storage on the water state (DSC) of dough samples.

Storage Time	Sample	Tm, Onset (°C)	ΔH (J/g)	FW (%)
Day 0	FD-control	−8.22 ± 0.13 ^c^	36.64 ± 0.56 ^a^	24.38 ± 0.37 ^a^
	FD-alc	−6.96 ± 0.29 ^a^	31.77 ± 0.15 ^b^	21.90 ± 0.10 ^b^
	FD-sav	−7.60 ± 0.22 ^b^	29.73 ± 0.77 ^c^	19.73 ± 0.5 ^c^
	FD-alc+sav	−6.57 ± 0.19 ^a^	37.33 ± 1.75 ^a^	24.56 ± 1.15 ^a^
Day 30	FRD-control	−6.25 ± 0.37 ^a^	51.26 ± 2.22 ^a^	33.96 ± 1.47 ^a^
	FRD-alc	−7.33 ± 0.28 ^b^	40.15 ± 1.39 ^b^	27.47 ± 0.95 ^b^
	FRD-sav	−5.86 ± 0.29 ^a^	36.42 ± 1.43 ^b^	24.91 ± 0.98 ^c^
	FRD-alc+sav	−7.38 ± 0.37 ^b^	38.59 ± 0.39 ^ab^	26.87 ± 0.27 ^b^

Same letters at the same colon show no significant differences at the same frozen storage time (*p* > 0.05). FD-control: fresh control dough; FD-alc: fresh dough containing FGH treated with alcalase; FD-sav: fresh dough containing FGH treated with savinase; FD-alc+sav: fresh dough containing FGH treated with alcalase + FGH treated with savinase; FRD-control: frozen control dough; FRD-alc: frozen dough containing FGH treated with alcalase; FRD-sav: frozen dough containing FGH treated with savinase; FRD-alc+sav: frozen dough containing FGH treated with alcalase + FGH treated with savinase.

**Table 3 foods-13-00139-t003:** Color properties of the bread samples.

Storage Time	Samples	Crust				Crumb			
		L*	a*	b*	ΔE	L*	a*	b*	ΔE
Day 0	Control-B	62.67 ± 0.95 ^aA^	9.28 ± 0.63 ^bBC^	27.68 ± 0.93 ^aA^	-	65.17 ± 0.91 ^bC^	0.60 ± 0.20 ^aA^	14.61 ± 0.84 ^bC^	-
	Alc-B	55.00 ± 1.17 ^bC^	12.65 ± 0.74 ^aA^	22.67 ± 1.18 ^cC^	9.79 ± 1.57 ^aA^	65.59 ± 1.14 ^bBC^	0.48 ± 0.27 ^aAB^	14.74 ± 0.96 ^bC^	1.23 ± 0.85 ^bB^
	Sav-B	55.83 ± 1.13 ^bC^	12.87 ± 0.67 ^aA^	24.67 ± 1.36b ^BC^	8.33 ± 1.59 ^aA^	68.05 ± 1.06 ^aA^	0.36 ± 0.17 ^aABC^	15.75 ± 0.53 ^abBC^	3.15 ± 1.03 ^aA^
	Alc+Sav-B	55.49 ± 1.82 ^bC^	13.05 ± 0.37 ^aA^	27.07 ± 1.39 ^aA^	8.25 ± 1.74 ^aA^	67.72 ± 1.02 ^aAB^	0.28 ± 0.17 ^aBC^	16.30 ± 0.41 ^aB^	3.16 ± 0.73 ^aA^
Day 30	Control-B	62.61 ± 0.66 ^aA^	8.44 ± 0.36 ^bC^	27.08 ± 0.69 ^aA^	1.30 ± 0.52 ^bC^	68.33 ± 1.58 ^aA^	0.16 ± 0.01 ^aC^	16.89 ± 0.21 ^abAB^	4.07 ± 0.96 ^aA^
	Alc-B	61.51 ± 2.18 ^aA^	9.98 ± 1.11 ^abBC^	26.94 ± 0.85 ^aA^	2.13 ± 2.03 ^bC^	67.24 ± 1.69 ^aABC^	0.15 ± 0.08 ^aC^	16.73 ± 0.37 ^bAB^	3.17 ± 1.25 ^aA^
	Sav-B	60.83 ± 1.34a ^AB^	9.40 ± 1.11 ^abBC^	25.82 ± 0.80 ^aAB^	2.92 ± 1.27 ^abBC^	68.26 ± 0.71 ^aA^	0.23 ± 0.11 ^aBC^	17.71 ± 0.71 ^aA^	4.39 ± 0.92 ^aA^
	Alc+Sav-B	58.58 ± 0.70 ^bB^	10.48 ± 1.31 ^aB^	26.08 ± 1.29 ^aAB^	4.81 ± 0.97 ^aB^	68.59 ± 0.88 ^aA^	0.11 ± 0.13 ^aC^	17.54 ± 0.70 ^abA^	4.58 ± 0.74 ^aA^

Data are expressed as mean ± standard deviation of triplicate determinations. Means with different lowercase letters in the same column indicate significant differences (*p* < 0.05) for the samples at the same storage time. Different uppercase letters show statistically significant effects of frozen storage (*p* < 0.05). Control-B: control bread; Alc-B: bread containing FGH treated with alcalase; Sav-B: bread containing FGH treated with savinase; Alc+Sav-B: bread containing both FGH. Same letters show no significant differences at the same frozen storage time.

## Data Availability

Data is contained within the article.

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
