# Peer review of "Effects of Fish Skin Gelatin Hydrolysates Treated with Alcalase and Savinase on Frozen Dough and Bread Quality"

_foods, 2023, doi:10.3390/foods13010139_

Round 1
Reviewer 1 Report
Comments and Suggestions for Authors
Dear Authors,
the reviewed manuscript concerns an important issue: improving the quality of bread from the postponed baking. The authors propose an interesting solution, which is the use of an additive based on a by-product from fish processing that has been enzymatically processed. The scope of the research and the discussion of the results are correct. Comments on the text of the manuscript, including supplementary discussion of the results, are presented in the file.
The results of farinographic analyzes of the control dough (control recipe) and test doughs (doughs in which 1% of the flour mass was replaced by the tested additive) would also be valuable. Maybe even tested before and after freezing/thawing. Please respond to this comment.
By far the greatest weakness of this study is the omission of the organoleptic assessment. Bread was subjected to the instrumental tests for quality characteristics (loaf volume, crumb hardness, crust and crumb colour), which are also subject to consumer evaluation, but this is the key one. Organoleptic assessment performed by an appropriately trained panel is an extremely important part of the comprehensive assessment of the finished product, especially when striving to improve quality. Please respond to this comment.
In this study, it is also possible to assess the relationship between the oragnoleptic quality characteristics measured by instrumental methods and the results of the organoleptic assessment performed by the panelists.
Kind regards
Reviewer

Author Response
We really thank you for your valuable recommendations and corrections. They were so beneficial to improve our manuscript. We made all corrections and highlighted them in the manuscript. We also responded your questions at the below. Hopefully, our updated manuscript will be accepted after these corrections and revisions.

Reviewer 2 Report
Comments and Suggestions for Authors
1) Line 96 Please specify how the water absorption value was determined.
2) Line 144 Please explain the SDS acronym.
3) Line 171 Please insert the city of manufacturer.
4) Line 181 Please express speed 4 in rpm.
5) Line 185 Please insert the city of manufacturer.
6) Line 186 Was humidification used during baking?
7) Line 194 Please insert the city/state of manufacturer.
8) Line 200 Please replace the X sign with the multiplication sign.
9) Line 225 For a complete TPA characterization, the authors need to determine all texture parameters: hardness, firmness, springiness and cohesiveness. Why was only hardness chosen?
10) What is the cost of FGH adding to the recipe?
11) I recommend to the authors to include in the paper the TPA results for 2, 3 and 4 days after baking.
12) What is the risk of consuming bread with FGH by consumers with fish allergy?
Author Response

(The authors gave the same response as above.)

Reviewer 3 Report
Comments and Suggestions for Authors
Ms. Ref. No.: Foods 2777879
Effects of Fish Skin Gelatin Hydrolysates Treated with Alcalase and Savinase on Frozen Dough and Bread Quality
The objective of the work is to study the effect of reuse of fish gelatin in the technological properties of frozen dough and bread quality. The work has a perfectly defined objective. And the results are well explained, establishing the conclusion that the hydrolyzate (FGH treated with alc) is what allows improving the characteristics of frozen bread.
The methodology is generally well defined. However, the presentation of the data needs to be improved.
In the manuscript minor changes must be considered.
In the Results section:
In Figure 2. the image is blurred.In Figure 4. should include the DS and better group the samples. That is, group the samples so that they could be better understood. (FD-Control; FD-Sav; FD-Alc; FD-Alc+Sav) (FDR-Control; FDR-Sav; FDR-Alc; FDR-Alc+Sav). In addition, a statistical analysis of the data must be included to establish the differences between the samples.
In Figure 5. should be change spesific to specific.
Author Response
We really thank you for your valuable contributions. We revised the manuscript based on your recommendations.

Round 2
Reviewer 1 Report
Comments and Suggestions for Authors
Dear Authors,
Thank you for preparing a response to my comments. In most of them I say that they have been taken into account, while in the rest I have added explanations.
Kind regards
Reviewer